# Bioaccessible Organosulfur Compounds in Broccoli Stalks Modulate the Inflammatory Mediators Involved in Inflammatory Bowel Disease

**DOI:** 10.3390/ijms25020800

**Published:** 2024-01-08

**Authors:** Antonio Costa-Pérez, Paola Sánchez-Bravo, Sonia Medina, Raúl Domínguez-Perles, Cristina García-Viguera

**Affiliations:** 1Laboratorio de Fitoquímica y Alimentos Saludables (LabFAS), CEBAS-CSIC, Espinardo, 30100 Murcia, Spain; acosta@cebas.csic.es (A.C.-P.); paola.sanchezb@umh.es (P.S.-B.); smescudero@cebas.csic.es (S.M.); cgviguera@cebas.csic.es (C.G.-V.); 2Centro de Investigación e Innovación Agroalimentaria y Agroambiental (CIAGRO), Universidad Miguel Hernández de Elche (UMH), Carretera de Beniel km 3.2, 03312 Orihuela, Alicante, Spain

**Keywords:** broccoli byproducts, glucosinolates, isothiocyanates, interleukins and cytokines, signalling, inflammation

## Abstract

Inflammatory diseases are strongly associated with global morbidity and mortality. Several mediators are involved in this process, including proinflammatory interleukins and cytokines produced by damaged tissues that, somehow, act as initiators of the autoreactive immune response. Bioactive compounds present in plant-based foods and byproducts have been largely considered active agents with the potential to treat or prevent inflammatory diseases, being a valuable alternative to traditional therapeutic agents used nowadays, which present several side effects. In this regard, the present research uncovers the anti-inflammatory activity of the bioaccessible fraction of broccoli stalks processed, by applying different conditions that render specific concentrations of bioactive sulforaphane (SFN). The raw materials’ extracts exhibited significantly different contents of total glucosinolates (GSLs) that ranged between 3993.29 and 12,296.48 mg/kg dry weight (dw), with glucoraphanin as the most abundant one, followed by GI and GE. The indolic GSLs were represented by hydroxy-glucobrassicin, glucobrassicin, methoxy-glucobrassicin, and neo-glucobrassicin, with the two latter as the most abundant. Additionally, SFN and indole-3-carbinol were found in lower concentrations than the corresponding GSL precursors in the raw materials. When exploring the bioaccessibility of these organosulfur compounds, the GSL of all matrices remained at levels lower than the limit of detection, while SFN was the only breakdown product that remained stable and at quantifiable concentrations. The highest concentration of bioaccessible SFN was provided by the high-ITC materials (~4.00 mg/kg dw). The results retrieved on the cytotoxicity of the referred extracts evidenced that the range of supplementation of growth media tested (0.002–430.400 µg of organosulfur compounds/mL) did not display cytotoxic effects on Caco-2 cells. The obtained extracts were assessed based on their capacity to reduce the production of key proinflammatory cytokines (interleukin 6 (IL-6), IL-8, and TNF-α) by the intestinal epithelium. Most of the tested processing conditions provided plant material with significant anti-inflammatory activity and the absence of cytotoxic effects. These data confirm that SFN from broccoli stalks, processed to optimize the bioaccessible concentration of SFN, may be potential therapeutic leads to treat or prevent human intestinal inflammation.

## 1. Introduction

Inflammatory bowel disease (IBD) is a chronic and relapsing disorder affecting the gastrointestinal tract. This pathology is associated with a range of clinical phenotypes featured by specific traits, namely ulcerative colitis (UC) and Crohn’s disease (CD) [1]. The incidence and prevalence of IBD have augmented over the past decade due to the modification of lifestyles [2]. Alteration of the gut microbiota homeostasis, also known as dysbiosis, has also been recently identified as a critical factor involved in the pathogenesis of IBD. Indeed, current evidence regarding this inflammatory process has proved a perturbation of the gut microbiota not only in human IBD patients but also in mice colitis models. This evidence has been further supported by the success of gut microbiota-based therapeutic approaches. Nonetheless, currently, additional research is still (e.g., well-designed randomized control trials and mouse models) needed to understand the actual interaction between intestinal microbiota and the altered functioning of the immune system with autoreactive consequences [3]. In this regard, although more than 100 genes have been associated with an increased susceptibility [4], the genetic factor does not explain the epidemiology of IBD completely, suggesting that this is strongly influenced by environmental factors like societal changes, healthcare, air quality, or food [5].

Among the diverse environmental factors affecting IBD, the dietary pattern has been linked positively with health and longevity in the case of this disease, being a source of specific biomolecules, such as fatty acids, responsible for modulating the interactions between the gut microbiome and immune cells [6], or dietary fibre that is metabolized by the intestinal microbiota into short-chain fatty acids, contributing to the prevention of the transcription of proinflammatory mediators [7]. However, beyond nutrients, other bioactive compounds present in plant-based foods have been also associated with protection against inflammation [8,9].

Concerning the plant-based food sources of anti-inflammatory bioactive compounds, broccoli (*Brassica oleracea* var. italica) has been noticed because of its phytochemical and nutritional wealth, including organosulfur compounds [10]. This family of phytochemicals involves glucosinolates (GSLs), whose chemical structure is featured by the presence of a β-D-thioglucose with a sulfonated oxime moiety and a variable side chain (R) derived from specific amino acids that allow the classification of GSLs into the aliphatic, indolic, and aromatic classes [11]. Nonetheless, the physiological (bioactive) meaning of GSL is based on their role as precursors of the bioactive derivatives, namely isothiocyanates (ITC), nitriles, thiocyanates, epithionitriles, and oxazolidine-2-thions, which are differentially formed depending on the microenvironment’s physicochemical conditions [12], using enzymatic reactions catalyzed by a β–thioglucosidase (mirosynase) [13,14,15]. According to the chemical structure and the subsequent capacity to react with key molecules involved in the inflammatory process, these compounds have been associated with anti-inflammatory power by using diverse in vitro, in vivo, and in silico research [16,17]. Due to this composition, broccoli materials could represent good sources of functional ingredients against inflammation [10,18,19].

Interestingly, these compounds are found not only in edible broccoli inflorescences but also in related byproducts (leaves and stalks) that could represent up to 70% of the aerial parts of broccoli plants [19]. Therefore, the occurrence of these valuable bioactive phytochemicals in agro-food byproducts, which represent an environmental constraint that needs new valorization alternatives to reduce their impact on local environments and enhance the sustainability of agro-food production, allows the envisaging of new eco-friendly applications for the design and development of new functional foods [20]. To achieve this goal, the adjustment of the processing conditions towards coproducts displaying the highest ITC content possible has been previously addressed by our research group, leading to obtaining stable and safe materials with different profiles of organosulfur compounds [21]. Nonetheless, the phytochemical burden of the stabilized materials retrieved cannot be the only criteria for decision-making on the best processing conditions and phytochemical profile, because the effects of gastrointestinal digestion in the delivery of the active fraction and their actual anti-inflammatory power remain essential. In this regard, to date, the beneficial effect of ITC on IBD’s risk factors has been demonstrated based on the extracts of raw materials, while the bioaccessible fraction obtained after simulated gastrointestinal digestion, which varies significantly relative to the former, remains underexplored.

In this context, the bioactive breakdown products of GSL have been demonstrated as bioaccessible [22,23], reaching even higher concentrations as a result of gastrointestinal digestion in comparison with the plant material. Recent research has put the spotlight on the effect of alternative processing conditions to obtain the highest concentrations of bioaccessible ITC [21]. Nevertheless, it must go further and investigate their capacity to prevent the inflammatory cascade in IBD. Based on these antecedents, this study pursues the assessment of bioaccessible ITC and indoles profiles compared to the concentrations present in the raw material (analytical extract), and their anti-inflammatory activity in vitro, in relation to intestinal epithelial cells, by modulating the expression of proinflammatory cytokines (IL-6, IL-8, and TNF-α) secreted by cells exposed to a proinflammatory stimulus linked to intestinal inflammation (IL-1β).

## 2. Results

### 2.1. Quantitative Organosulfur Compound Content of Stabilized Broccoli Stalks

To establish the capacity to prevent intestinal inflammation of the raw material extracts and bioaccessible fractions of broccoli stalks, they were processed specifically to obtain high-GSL, high-ITC, and GSL/ITC balanced materials, according to the conditions described in the literature [21]. This approach evidenced that the quantitative profile of organosulfur compounds of broccoli stalks included aliphatic (glucoiberin (GI), glucoraphanin (GR), glucoerucin (GE), and aromatic (gluconasturtiin (PE)), and indolic (hydroxy-glucobrassicin (HGB), glucobrassicin (GB), methoxy-glucobrassicin (MGB), and neo-glucobrassicin (NGB)) glucosinolates (Table 1).

The analysis of the quantitative organosulfur profile of the three broccoli stalks’ raw materials (high-GSL, high-ITC, and GSL/ITC balanced contents) resulted in the total contents of GSL of 12,296.48, 4408.71, and 3993.29 mg/kg dry weight (dw) being detected, respectively. For aliphatic GSLs, GR was the most abundant GSL in broccoli stalks as has already been described for several broccoli materials [24]. This GSL also exhibited the most significant difference among materials, being its concentration in high-GSL materials around three-fold higher in comparison with high-ITC and GSL/ITC, on average.

Apart from GR, additional aliphatic GSLs monitored in broccoli stalks displayed a similar distribution among the three types of broccoli stalks according to their content of organosulfur compounds. Thus, GI and GE displayed similar concentrations in high-GSL materials (1227.78 mg/kg dw, on average) (Table 1). In this regard, the GE content of high-GSL broccoli stalks was 4.7- and 8.3-fold higher than in the high-ITC and GSL/ITC balanced materials, respectively. Concerning the GI, its concentration ranged between 241.46 and 1347.44 mg/kg dw in the three broccoli stalk materials, which emphasized high-GSL material as being the most significant source of this GSL (Table 1).

The indolic GSLs were represented by HGB, GB, MGB, and NGB. The two latter were the most abundant in the three matrices (Table 1). Hence, while MGB was found at the highest concentration in high-GSL and high-ITC materials (Table 1), no differences were noted among them. Regarding NGB, no significantly different concentrations were observed between high-GSL and GSL/ITC balanced matrices, showing that high-ITC materials had the lowest concentration for this indole GSL (Table 1). The amounts of the remaining indole GSLs (GB and HGB) were 192.05 and 78.24 mg/kg dw, on average, respectively (Table 1). Once again, the best source varied depending on the GSLs considered. While the best source for HGB was the high-ITC matrix, no significant differences in the GB content were observed.

Additionally, this study enabled the detection and quantification of the aromatic GSL PE. For this compound, the high-GSL material exhibited the highest concentration, which was surpassed by more than three-fold the burden recorded in high-ITC and GSL/ITC balanced materials (Table 1).

Regarding the ITC and indoles, it was found that SFN and I3C had lower contents than the corresponding GSL precursors (Table 1). In this context, SFN occurred in greater quantity in the GSL/ITC balanced material, surpassing by four times the concentration recorded in high-GSL and high-ITC materials (Table 1). Similarly, the highest concentration of I3C was found in GSL/ITC balanced and high-ITC materials (Table 1).

### 2.2. Effect of Gastrointestinal Digestion on the Bioaccessibility of Organosulfur Compounds

Once the contrasting values of processed broccoli stalks as a source of GSLs and their breakdown products were obtained, the extent to which gastrointestinal digestion renders ITC and indoles from the broccoli stalk to the intestinal lumen for their ulterior absorption (bioaccessibility) was explored; this is of pivotal importance to understand the biological relevance.

Related to this, the GSL content of the high-GSL, high-ITC, and GSL/ITC balanced materials was at concentrations lower than the limit of detection of the analytical technique. On the other hand, regarding the findings on the ITC and indoles’ bioaccessibility observed in this study, SFN was the only ITC that remained stable and at concentrations higher than the limit of quantification (>LOQ) in the digestion products (Figure 1).

As expected, the material providing the highest concentration of bioactive SFN in the bioaccessible fraction, after in vitro simulation of gastrointestinal digestion, was the high-ITC materials (~4.00 mg/kg dw), which surpassed the concentration provided by the high-GSL and GSL/ITC balanced materials by 31.9% and 73.7%, respectively (Figure 1).

### 2.3. Cytotoxicity of the Analytical Extracts and Digested Products in Intestinal Epithelium Cells

To establish the ability of the separate analytical extracts and digestion products to modulate the cytokine profile, firstly, a viability analysis was carried out using a human colon adenocarcinoma cell line (Caco-2). For this purpose, the cytotoxicity of raw broccoli stalk extracts (Figure 2A) and the digestion products (Figure 2B) of high-GSL, high-ITC, and GSL/ITC balanced materials was assessed by exposing Caco-2 cells to rising percentages of supplementation of the growth media from 5.0% to 35.0% that provided a concentration of total organosulfur compounds that ranged between 20.0 and 430 µg/mL of culture media concerning the extracts of raw material and for the bioaccessible ITC between 0.002 and 0.025 µg/mL of culture media (final concentrations) (Figure 2).

The results proved that none of the supplementation percentages with raw material extracts or digestion products had cytotoxic effects on Caco-2 cells (Figure 2). So, in all cases, cell viability remained almost constant at around ~100%, without significantly different percentages among the five supplementation levels tested (5.0% (20.0–61.5 µg of organosulfur compounds/mL for raw material extracts and 0.002–0.004 µg of organosulfur compounds/mL for bioaccessible fractions), 7.5% (29.9–92.2 µg of organosulfur compounds/mL for raw material extracts and 0.002–0.005 µg of organosulfur compounds/mL for bioaccessible fractions), 10.0% (39.9-123.0 µg of organosulfur compounds/mL for raw material extracts and 0.003–0.007 µg of organosulfur compounds/mL for bioaccessible fractions), 20.0% (79.9–245.9 µg of organosulfur compounds/mL raw material extracts and 0.006–0.014 µg of organosulfur compounds/mL for bioaccessible fractions), and 35.0% (139.8–430.4 µg of organosulfur compounds/mL for raw material extracts and 0.011–0.025 µg of organosulfur compounds/mL for bioaccessible fractions)). This analysis allowed the establishment of the supplementation percentage to evaluate the capacity of broccoli stalks to modulate the inflammatory mediators involved in IBD.

### 2.4. Anti-Inflammatory Activity of Broccoli Stalks’ Analytical Extracts and Bioaccessible Fractions

Interleukins are a group of low-molecular-weight proteins and glycoproteins, produced by various types of cells, which act by mediating cellular interactions, contributing to regulating the immune response and the course of inflammation [25]. At the intestinal level, interleukins intervene as growth factors and mediators of soluble proteins essential for intercellular communication of the intestinal mucosa, factors for the maintenance of homeostasis, as well as key participants in intestinal inflammation and associated damage [26]. To determine the extent to which organosulfur compounds are competent to modulate the cytokine level in a proinflammatory environment, Caco-2 cells were stimulated with IL-1β (25 ng/mL), thus triggering the inflammatory phenotype in the intestinal epithelial cells [27]. For this purpose, changes in the concentration of three key proinflammatory cytokines (IL-6, IL-8, and TNF-α) after supplementation of raw material extracts and bioaccessible fractions of broccoli stalks were evaluated (Figure 3).

The IL-6 was not detected in untreated samples (negative control without either broccoli stalk extracts or proinflammatory stimulus). In contrast, positive controls represented by Caco-2 cells exposed to a proinflammatory stimulus (25 ng/mL of IL-1β) provided high levels of IL-6 (around 60.00 pg/mL) (Figure 3). Concerning the ability of the analytical extracts of raw broccoli stalk to modulate the level of IL-6, high-GSL and GSL/ITC balanced samples lowered significantly the concentration of this interleukin by almost three times, up to ~20.00 pg/mL. Conversely, the extract of the high-ITC broccoli stalk material induced a less intense (although still statistically significant) decrease in IL-6 to reach a final average concentration of 39.90 pg/mL (Figure 3). Interestingly, the bioaccessible fraction of high-GSL, high-ITC, and GSL/ITC balanced materials inhibited significantly the secretion of IL-6 by the intestinal epithelial cells by up to ~20.00 pg/mL (Figure 3), which is consistent with the anti-inflammatory capacity of ITC.

When analyzing the capacity of organosulfur compounds to decrease the level of the pro-inflammatory and chemotactic interleukin (chemokine) IL-8 secreted by intestinal epithelial cells as a result of the exposure to IL-1β, it was observed that the level recorded in the cells only exposed to the pro-inflammatory stimulus 354.84 pg/mL was significantly inhibited by the organosulfur compounds contained in the extracts of high-GSL and GSL/ITC balanced raw materials by 68.0% (15.6%–99.2%), on average (Figure 3). Moreover, it is important to point out that the analytical extract of broccoli material with a high ITC content was not able to reduce IL-8 levels (338.90 pg/mL), showing no statistical differences with positive control samples. On the other hand, concerning the bioaccessible fraction of the different broccoli stalk materials, the anti-inflammatory power expressed as the capacity to inhibit the secretion of IL-8 by intestinal epithelial cells for high-GSL, high-ITC, and GSL/ITC balanced samples enabled a significant reduction of up to 218.36, 191.02, and 109.63 pg/mL, respectively, relative to the positive control (Figure 3).

Finally, concerning the pro-inflammatory interleukin, TNF-α, extracts of raw broccoli stalk materials (high-GSL, high-ITC, and GSL/ITC balanced contents) were able to reduce TNF-α production (41.90, 35.60, and 39.30 pg/mL, on average, respectively) compared to positive control samples (51.30, pg/mL, on average) although statistical significance was not reached due to the variability of the data (Figure 3). On the other hand, the digestion products of all three broccoli stalk materials significantly lowered the TNF-α production recorded in Caco-2 cells treated only with pro-inflammatory IL-1β (16.10, 10.60, and 21.90 pg/mL, respectively, for high-GSL, high-ITC, and GSL/ITC balanced samples to similar levels to those observed in basal conditions (negative control samples) (9.20 pg/mL on average) (Figure 3).

### 2.5. Principal Component Analysis

To determine the relationship between the inflammatory compounds analyzed in broccoli stalks (raw material extracts and bioaccessible fractions), PCA was applied to the experimental results. The PCA performed on the data matrix that included the concentration of GSLs and ITC in the separated matrices, as well as the concentration of inflammation-signalling interleukins (IL-6, IL-8, and TNF-α) revealed acceptable scores for the two principal components (PC1 and PC2), which explained 73.2% of the variance (Figure 4A).

In addition, PCA enabled the reduction in the data dimension, showing the clustering into two main groups, coloured (raw material extracts) and noncoloured (products of the gastrointestinal digestions). The joint analysis of the biplot results confirmed that the raw material extracts were characterized by the content of GSLs (GR, GE, HGB, GB, MGB, NGB, PE and I3C) while the gastrointestinal products lost these compounds and the content with a high concentration of ITC SFN. Also, concerning the inflammatory mediators, both raw material extracts and digestion products correlated negatively with the presence of proinflammatory IL-6 and the chemotactic factor, IL-8, while only the digestion products were competent to reduce the concentration of the proinflammatory cytokine, TNF-α, as a result of their concentration of bioactive SFN (Figure 4B).

## 3. Discussion

In regard to IBD, the anti-inflammatory power of organosulfur compounds present in broccoli stalks needs to be ascertained by providing evidence of the remaining anti-inflammatory capacity after digestion, according to the bioaccessibility of bioactive SFN. This biological function is mainly developed via modulating the cellular signalling responsible for the acquired immune response by the bioactive compounds present in the digestion products of broccoli stalks that would support the development of new valorization alternatives [20,21]. This information is required to establish the capacity of anti-inflammatory compounds, already described in brassica foods and byproducts (ITC) [28], to prevent the initiation of the inflammatory cascade triggered in epithelial cells as a result of the exposure to proinflammatory agents in the intestine [29]. To test this hypothesis, comparison was made of the quantitative organosulfur content of broccoli stalks processed according to Costa et al. (2022), to obtain different profiles of anti-inflammatory phytochemicals (GSL, ITC, and indoles) in the raw materials relative to the digestion products [21]. In this regard, the present work confirmed previous results on the association of the preprocessing and dehydration conditions with the efficiency of the myrosinase enzyme responsible for GSL breakdown into bioactive ITC upon physical disruption of the tissue [30].

The different dehydration conditions applied in the present work provided quantitative profiles of organosulfur compounds in broccoli stalks that agree with previous descriptions in the literature, including aliphatic (GR, GE, and GI), aromatic (PE), and indolic (HGB, GB, MGB, and NGB) GSLs [10,18,31]. Also, concerning GSLs, GR was the most abundant GSL in broccoli stalks, which is in good agreement with previous reports in the literature [24]. Hence, although the concentration of GR in the different materials obtained from raw broccoli stalks remained at a lower average level than that reported for inflorescences [32], the concentration obtained highlighted a powerful biological capacity and health-promoting factors [16]. Apart from GR, the level of GE is especially relevant because of its hydrolysis into the corresponding ITC derivative (E), which is interconverted to a large extent into SFN (the only ITC present in the bioaccessible fraction), with a direct impact on the biological power of the plant materials under consideration [28].

As referred to before, the formation of SFN is a consequence of GR catabolism mediated by the enzyme, myrosinase [33], but also as a result of the E/SFN interconversion [34], which is an important source of SFN because of the abundance of GE in broccoli stalks [18]. These results, in conjunction with the broad description in the literature about the anti-inflammatory potential of SFN [16,35,36], further support the interest in broccoli stalks as a source of bioactive ITC with the capability to modulate the inflammatory and immunological mediators that constitute an essential mechanism for the progression of the intestinal inflammatory process [29]. Even more, the presence of GI should be positively evaluated because of its role as the precursor of IB, an ITC associated with a powerful anti-inflammatory activity via modulation of inducible nitric oxide (NO) synthase (iNOS) (key for the production of NO from IL-arginine [37]), TNF-α, IL-1α, and activated NF-κB [38].

Alternatively, the indolic GSL profile of the materials characterized in the present work was represented by HGB, GB, MGB, and NGB, with MGB and NGB being the most abundant. Thus, the concentrations determined in the broccoli stalks for indolic GSLs oscillate around average values very similar to previous studies [10,18,39] that indicate that this GSL type would contribute to the anti-inflammatory and antitumoral attributions of brassica foods [40,41].

Despite the biological advantages deduced from the aliphatic GSLs of broccoli stalks and sources of the bioactive derivatives (ITC, nitriles, thiocyanates, epithionitriles, and oxazolidine-2-thions), the concentration of these anti-inflammatory compounds in the bioaccessible fraction appears to be quite limited because of the effect of the physicochemical conditions during digestion of GSLs and the activity of hydrolytic enzymes [12]. In this regard, the stability of the different GSLs and breakdown products during digestion seems to be dependent, not only on their specific chemical structure but also on the physical properties of the vegetable material, which could strongly make the effect of gastrointestinal digestion conditional on the release and stability of the organosulfur compounds [23,42]. In this regard, the metabolism of human microbiota provides valuable input regarding the hydrolysis of GSLs into their bioactive counterparts, ITC and indoles. In this regard, in the last decade, the capacity of the human microbiome to provide myrosinase activity has emerged as a significant mechanism that potentially can raise the beneficial effects of consumption of plant materials considered rich sources of dietary GSLs. Nonetheless, the diverse composition of the human microbiome in separate individuals/populations, in conjunction with the diversity of GSLs present in diets may lead to greater variability in the actual dose of prohealth compounds absorbed by the human body [43]. Interestingly, despite the diversity of GSLs in plant material, SFN was the only bioactive derivative detected in the digestion products [23]. Bioaccessible SFN is the result of GR catabolism hydrolysis [33], but also of E/SFN interconversion [34], which is an important source of SFN because of the abundance of GE in broccoli stalks [18]. Thus, this result, in conjunction with the broad description in the literature about the anti-inflammatory potential of SFN [16,35,36], further supports the interest in broccoli stalks as a functional ingredient in the management of disorders that involve inflammation, such as IBD [29]. This outcome is aligned with the previous report on the bioaccessibility of GLS breakdown products obtained via the digestion of cruciferous sprouts using the same methodology [23]. In this regard, the real anti-inflammatory activity of the ITC concentrations reached after gastrointestinal digestion in the intestinal lumen remains unexplored. Additional research is essential to identify the compounds produced after digestion, their concentration in the bioaccessible fraction, and their contribution to an anti-inflammatory effect on the intestinal epithelium.

From these results, the extent to which the mixtures of GSL, ITC, and indoles of raw material extracts and the bioaccessible could help to adjust the profile of relevant proinflammatory cytokines (IL-6, IL-8, and TNFα) secreted by intestinal epithelial cells exposed to a proinflammatory stimulus was addressed [29], and thus, control of migration and maturation of lamina propria resident macrophages [44]. Thus, organosulfur compounds may contribute to preventing the course of IBD by finetuning the key participants in intestinal inflammation, the associated damage, and immune response, and thereby the course of intestinal inflammation for intercellular communication of the intestinal mucosa, as key participants in intestinal inflammation and the associated damage [25,26]. In the present work, this immunomodulatory capacity was demonstrated in an intestine inflammatory model based on the exposure of Caco-2 cells to the proinflammatory factor, IL-1β, thus modifying the intestinal epithelial cells’ phenotype to an inflammatory one, featured by an altered secretion of IL-6, IL-8, and TNFα [29]. Among the proinflammatory stimuli responsible for triggering IBD, the interplay between the commensal microbiota and the intestine’s resident immune cells tunes essential functions that include multifold interactions. In this regard, the microbiome is responsible for activating essential components of the innate and adaptive immune system, which alters, to some extent, the role of the immune system in the maintenance of host–microbe symbiosis. As a result, imbalances in microbiota–immunity interactions in the proper scenario are believed to contribute to the pathogenesis of a multitude of immune-mediated disorders, namely IBD [45].

The significant reduction of the IL-6 secretion by epithelial cells under proinflammatory conditions has been associated with materials processed under specific conditions, providing different concentrations of SFN [23,46]. This seems to be so because their GSL content, as a consequence of the enzymatic activity and other physical–chemical factors involved in this physiological process [47], is hydrolyzed, rendering high concentrations of bioaccessible SFN with biological power to inhibit the IL-6 production triggered by the proinflammatory stimulus. However, the lack of differences between samples providing distinct concentrations of bioaccessible SFN could be a result of the saturation of the action mechanism [48]. Beyond this, the separate effects observed may be due to the effect of the processing methods on additional phytochemical compounds present in broccoli stalks, mainly flavanols and hydroxycinnamic acids, which have been described in high concentrations in cruciferous vegetables and are responsible for antioxidant and anti-inflammatory power [49]. Thus, this study confirmed that digestive extracts of materials with high GLS, high ITC, and balanced GSL/ITC contents, might be used for regulating proinflammatory cytokine, IL-6, and contributing to resolving inflammation in IBD pathogenesis.

In addition to IL-6, when analyzing the capacity of organosulfur compounds to decrease the level of the proinflammatory and chemokine, IL-8, and the interleukin, TNFα, secreted by intestinal epithelial cells, significant inhibition was observed by the organosulfur compounds. Considering the unequal concentrations of SFN in the different broccoli matrices after gastrointestinal digestion, these findings might also be explained by the hormetic potential of SFN, represented by its dose–response curve, as well as the determination of the quantitative threshold point for SFN to ensure better therapeutic outcomes. This might depend upon several parameters, such as the proinflammatory stimulus, inflammation mediators, and microenvironmental and physiological conditions, among other factors [50].

This information, once evaluated in relation to the capacity to modulate the level and effect of inflammatory mediators, will provide evidence on the actual relevance of dietary brassicas to reduce and modulate the incidence and clinical evolution of IBD patients, in agreement with previous evaluations [8]. Indeed, our data confirm the anti-inflammatory potency of SFN of broccoli byproducts, via an in vitro model of the intestinal barrier, which is in line with previous studies developed using complementary models, such as murine macrophages [51], and human macrophage-like cell models derived from HL-60 cells [52].

## 4. Materials and Methods

### 4.1. Chemicals and Reagents

The standards of GSLs, namely glucoiberin, glucoraphanin, hydroxy-glucobrassicin, glucoerucin, glucobrassicin, gluconasturtiin methoxy-glucobrassicin, and neo-glucobrassicin (GI, GR, HGB, GE, GB, PE, MGB, NGB, respectively), and the standards of ITC and indoles, namely *D,L*-sulforaphane *L*-cysteine, *D,L*-sulforaphane glutathione, erucin, *D,L*-sulforaphane-*N*-acetyl-*L*-cysteine, iberin, indole-3-carbinol, sulforaphane, and 3,4-diindolylmethane (SFN-CYS, SFN-GSH, E, SFN-NAC, IB, I3C, SFN, DIM, respectively) were obtained from Phytoplan GmbH (Heidelberg, Germany). Acetic acid, hydrochloric acid, and ammonium acetate were obtained from Panreac labs (Barcelona, Spain), and the solvents methanol and acetonitrile (LC-MS grade) were supplied by JT-Baker (Philipsburg, NJ, USA). Deionized water was purified using a Milli-Q system (Millipore, Bedford, MA, USA).

Trypsin-EDTA, Eagle’s minimum essential medium (EMEM), *L*-glutamine, foetal bovine serum (FBS), penicillin/streptomycin, and essential amino acids were purchased from ThermoFisher Scientific (Madrid, Spain). The flat bottom 96-well plates were obtained from Corning (New York, NY, USA). Trypan Blue was obtained from Sigma-Aldrich (St. Louis, MO, USA).

### 4.2. Plant Material Collection and Processing

Broccoli plants (*Brassica oleracea* var. italica) were of the cultivar “Parthenon” and obtained from CMS (Cytoplasmic Male Sterility) commercial seeds provided by SAKATA Seed Ibérica (Alicante, Spain). Broccoli plants were cultivated in the fall–winter cycle of 2022, in the experimental field of the CEBAS-CSIC “La Matanza” Experimental Farm (Santomera, Murcia, SE Spain; 38°6′14″ N, 1°1′59″ W), under a semiarid Mediterranean climate. Harvesting was performed when plants presented mature commercial flowering heads and their quality parameters corresponded to the “marketable” class (compactness, homogeneity of grain, absence of secondary heads, absence of minor leaves in the head, and the total absence of hollow stems). The inflorescences were separated from the stalks manually, and the latter were transferred to the laboratory for further processing. The period between sampling and processing was less than 4 h to avoid the degradation of compounds. Once in the lab, broccoli stalks were processed according to the conditions previously described to obtain materials with high GSL content, high ITC content, and intermediate GSL and ITC content, according to a previous study performed by our research team [21]. Briefly, different thermal treatments were applied to obtain materials with a phytochemical profile according to the previous description by Costa et al. (2022). Thus, the separate stalk materials were dehydrated by applying low-temperature oven drying (40 °C for 72 h) and a descendent temperature gradient (Temp initial (75 °C)—temp final (60 °C) over 10 h). As a result of the dehydration process, the plant materials achieved constant weight corresponding to losses in the range of 83.0%–98.3%. The dehydrated samples were ground into a fine powder for further processing and analysis.

### 4.3. Extraction of Glucosinolates and Breakdown Products from Raw Materials

Samples (100 mg) were homogenized in 1 mL of ethanol/deionized water (50:50, *v*/*v*) and extracted at 70 °C for 20 min following the procedure described previously. All the obtained extracts were centrifuged at 1520× *g* for 5 min, filtered through a 0.22 µm PVDF filter (Millipore, MA, USA), and kept at −20 °C until posterior chromatographic analysis.

### 4.4. Simulated In Vitro Gastrointestinal Digestion

Simulated gastrointestinal digestion was performed on brassica stalk powder following the static in vitro gastrointestinal digestion methodology described in the literature [53,54], with minor modifications [22,23], applying simulated gastric fluid (SGF, Appendix A, Table A1) and simulated intestinal fluid (SIF, Appendix A, Table A1) in combinations of time and temperature according to the original methodology under continuous stirring. The bioaccessible fraction was filtered through a 0.22 µm PVDF filter (Millipore, MA, USA) and assessed based on their content of GSLs, ITC, and indoles using the HPLC-PAD-ESI-MSn and UHPLC-ESI-QqQ-MS/MS analytical methods.

### 4.5. HPLC-PAD-ESI-MSn Analysis of Glucosinolates

The chromatographic separation and spectrometry analysis of the individual GSLs present in the analytical extracts and gastrointestinal digestion products was performed using the methodology already described in the literature [23,55]. The identification of the target analytes was based on the retention time (min), parent ions, and fragmentation patterns compared to the authentic standards (Appendix A, Table A2). The quantification was performed on chromatograms recorded with a wavelength of 227 nm, applying calibration curves freshly prepared on each day of analysis, and the concentration was expressed as milligram per kilogram of dry weight (mg/kg dw).

### 4.6. UHPLC-ESI-QqQ-MS/MS Analysis of Isothiocyanate and Indole Derivatives

The chromatographic separation of ITC and indoles present in the extracts of raw materials and gastrointestinal digestion products was performed according to the methodology described previously [23,56,57]. The identification and quantification of ITC and indoles were based on their retention time, as well as the parental mass and specific fragmentation pattern monitored through qualitative and quantitative transitions (Appendix A, Table A3). The concentration of the compounds identified was calculated using standard curves freshly prepared on each day of analysis, and the concentration was expressed as mg/kg dw.

### 4.7. Cell Line, Culture Conditions, and Development of a Monolayer Intestinal Barrier

The colorectal Caco-2 (ATCC^®^HTB37) human cell line was obtained from the American Type Culture Collection (ATCC, Rockville, MD, USA). Cells were grown in EMEM, supplemented with 2 mM L-glutamine, 10% foetal bovine serum (FBS), and 1% non-essential amino acids at 37 °C in a humidified atmosphere containing 5% CO_2_. The passage number of the Caco-2 cells used in this study ranged from 16 to 18. The differentiation of Caco-2 cells into ciliated intestinal epithelium was achieved as reported in the literature [58]. The cell monolayer’s integrity was determined at transepithelial electrical resistances (TEER) higher than 600 Ω/cm^2^, according to the commercial measurement system, Millicell ERS (Millipore Co., Bedford, MA), calculated using the following equation: “*TEER* = (*R* − *Rb*) × *A*”, where “R” is the electrical resistance of the filter insert with the cell layer, “Rb” is the resistance of the filter alone, and “A” is the growth area of the filter in cm^2^. Once confluent, seeded Caco-2 cells were allowed to differentiate into ciliated cells for 21 days before the experiments, replacing the culture medium every 48–72 h.

### 4.8. Trypan Blue-Based Viability Assay

For the Trypan blue cytotoxicity assay, Caco-2 cells differentiated into ciliated cells were treated with a medium supplemented with broccoli stalk analytical extracts and gastrointestinal digestion products at 5.0% (20.0–61.5 µg of organosulfur compounds/mL for raw material extracts and 0.002–0.004 µg of organosulfur compounds/mL for bioaccessible fractions), 7.5% (29.9–92.2 µg of organosulfur compounds/mL for raw material extracts and 0.002–0.005 µg of organosulfur compounds/mL for bioaccessible fractions), 10.0% (39.9–123.0 µg of organosulfur compounds/mL for raw material extracts and 0.003–0.007 µg of organosulfur compounds/mL for bioaccessible fractions), 20.0% (79.9–245.9 µg of organosulfur compounds/mL for raw material extracts and 0.006–0.014 µg of organosulfur compounds/mL for bioaccessible fractions), and 35.0% (139.8–430.4 µg of organosulfur compounds/mL for raw material extracts and 0.011–0.025 µg of organosulfur compounds/mL for bioaccessible fractions)) (*n* = 3 for each treatment). Afterwards, cells were detached and stained with trypan blue. Cells with membrane integrity (excluding trypan blue) and those evidencing a lack of exclusion capacity (damaged cells) were counted, and the viability was expressed as the percentage of living cells after 24 h. Negative control cells received either the medium alone or the medium including the solvent used to prepare the target compounds’ dilutions.

### 4.9. Assessment of Inflammatory Modulators in the Monolayer Intestinal Barrier Model

Once the intestinal barrier model was set up, in order to assess the capacity of organosulfur compounds in broccoli stalks to prevent inflammation, confluent and ciliated cells were exposed to growth media supplemented with 30% broccoli stalk extracts (raw material extracts and digestion products) previously filtrated through a 0.22 µm PVDF filter (Millipore, MA, USA). After 1 h, 25 ng/mL of IL-1β (final concentration) as the proinflammatory stimulus was added to all wells (except for negative controls) for 10 h. Afterwards, the supernatants were collected, centrifuged at 2370× *g* for 10 min, and kept at −80 °C for further assessment of their content of cytokines (IL-6, IL-8, and TNF-α) following the manufacturer’s instructions (Abcam, Cambridge, UK). The limit of detection for the IL-6, IL-8, and TNFα immunoassay kits was 0.81, 12.30, and 4.32 pg/mL, respectively.

### 4.10. Statistical Analysis

All experimental conditions were performed in sextuplicate (*n* = 6). According to the normal distribution and homogeneity of variance of the data determined by Shapiro–Wilk (<50 samples) and Levene tests, respectively, the outcomes were subjected to a one-way analysis of variance (ANOVA) and, when statistical differences were identified at *p* < 0.05, the variables were compared using Tukey’s multiple range test.

The relationships between the concentration of bioactive phytochemicals in the separate samples and their capacity to protect against inflammation in Caco-2 cells were assessed by applying PCA. For this, the latter was applied as a pattern recognition unsupervised classification method, and the component score coefficient matrix, an output extracted by PCA, as well as Varimax and Kaiser normalization as the rotation method, enabled us to show the weight of the variables studied. Significant correlations were set at *p* < 0.05. All statistical analysis was performed using the SPSS program version 25.0 (SPSS Inc., Chicago, IL, USA).

## 5. Conclusions

Among all samples assessed (analytical extract and digestive products), a high bioaccessibility of ITCs from broccoli stalks, after simulated gastrointestinal digestion based on an in vitro static model, was confirmed. This fraction is responsible for strong anti-inflammatory activity due to its capacity to tackle the production of proinflammatory interleukins and chemokines, IL-6, IL-8, and TNF-α, by the intestinal epithelium in an inflammatory environment, contributing to adjusting the inflammation in IBD by finetuning factors closely involved in the development of the autoreactive immune response. Because of the strong correlation, SFN seems to be significantly involved in the successful development of this inhibitory activity. Accordingly, the use of broccoli byproducts as a functional ingredient in the agri-food industry would enrich the phytochemical composition of new foods, and their dietary application as adjuvants against chronic and degenerative inflammation diseases (especially affecting the intestinal tissues) is envisaged. Despite the new evidence provided in the present work, in the future, this needs to be complemented by additional mechanistic studies on the effect of an environment comprising specific interleukins and cytokines on the migration of macrophages to the inflammatory focus and their differentiation by developing specific pro- or anti-inflammatory phenotypes (M1 and M2, respectively), providing a specific tool of additional cytokines that trigger or modulate the cellular immune response due to their specific phenotype. Additional information on the prebiotic role of such bioaccessible fractions would help to understand the capacity of bioactive organosulfur compounds to act on the different aetiological agents and pathogenic mechanisms, thus providing evidence that will help to finetune the dietary advice for patients affected by IBD.

## Figures and Tables

**Figure 1 ijms-25-00800-f001:**
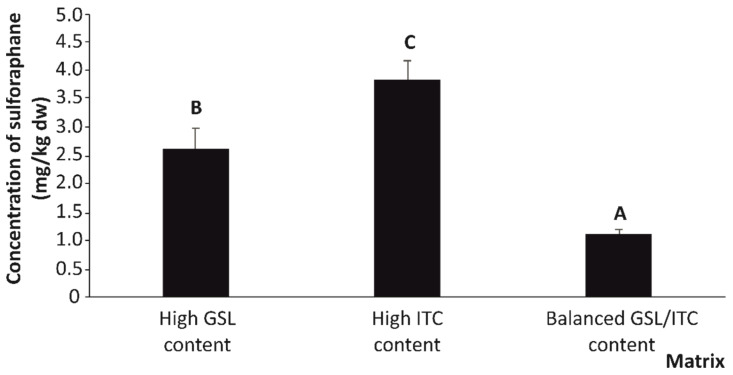
Concentration of bioaccessible sulforaphane (mg/kg dry weight (dw)) from the high-GSL, high-ITC, and GSL/ITC balanced materials (*n* = 6 per material). Bars with distinct letters are significantly different at *p* < 0.05 according to the one-way analysis of variance (ANOVA) and Tukey’s multiple range test.

**Figure 2 ijms-25-00800-f002:**
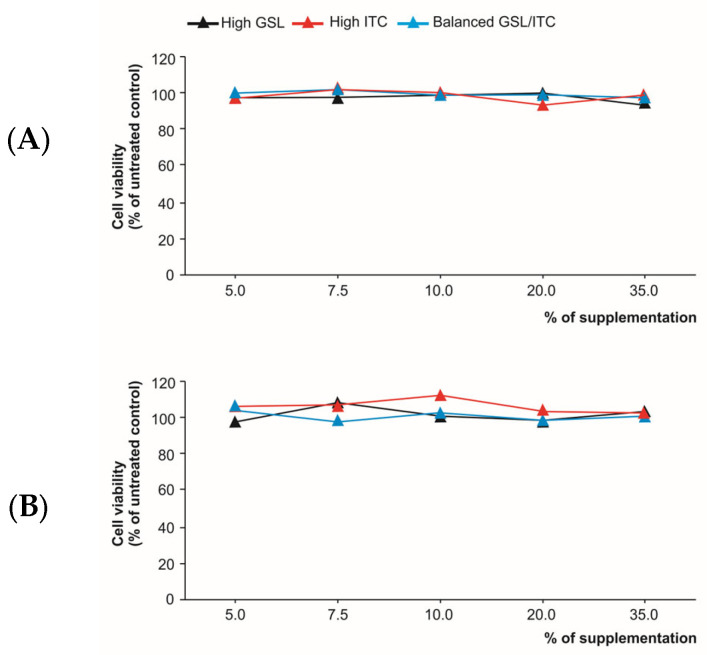
Trypan blue viability assay for the analysis of Caco-2 cells’ viability exposed to different percentages of supplementation of the growing media with GSL extracts of raw broccoli stalks (**A**) and the digestion products (**B**) (*n* = 3 for each treatment). High-GSL, high-ITC, and GSL/ITC balanced materials are represented by black, red, and blue lines, respectively.

**Figure 3 ijms-25-00800-f003:**
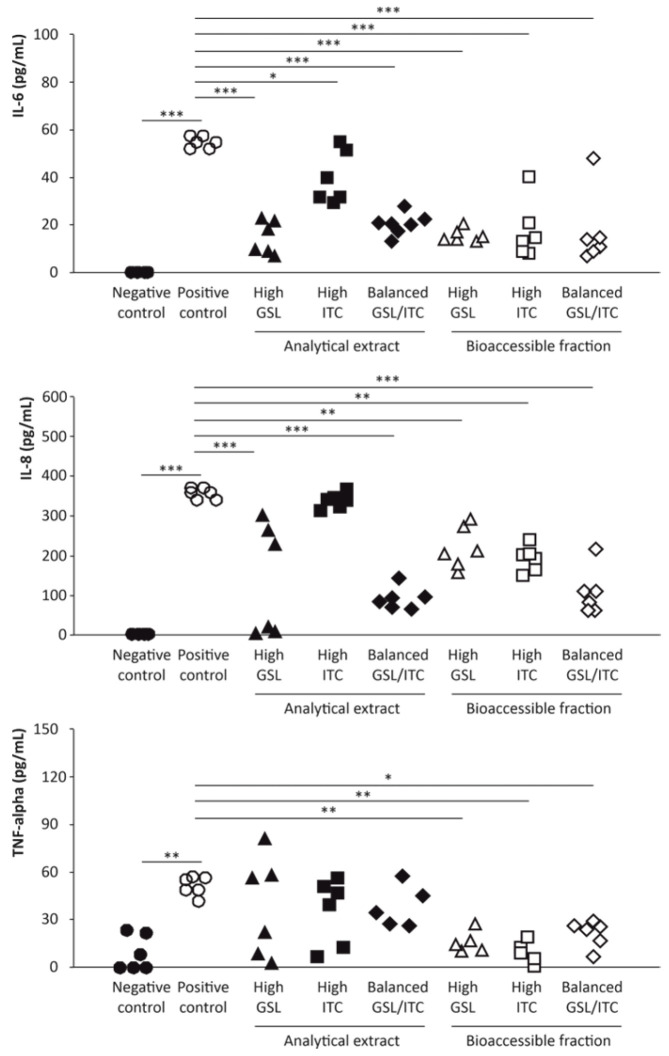
Dot plots of the quantitative profile of proinflammatory interleukins (IL-6, IL-8, and TNFα) induced by IL-1β in intestinal epithelial cells (Caco-2) mediated by the modulatory capacity of raw materials and digestive extracts of broccoli stalks processed to obtain high-glucosinolate, high-isothiocyanate, and glucosinolate/isothiocyanate balanced materials (high-GSL, high-ITC, and GSL/ITC balanced, respectively) (*n* = 6 for each treatment). Significant differences at *p* < 0.05 (*); *p* < 0.01 (**); and *p* < 0.001 (***) according to one-way analysis of variance (ANOVA) and Tukey’s multiple range test.

**Figure 4 ijms-25-00800-f004:**
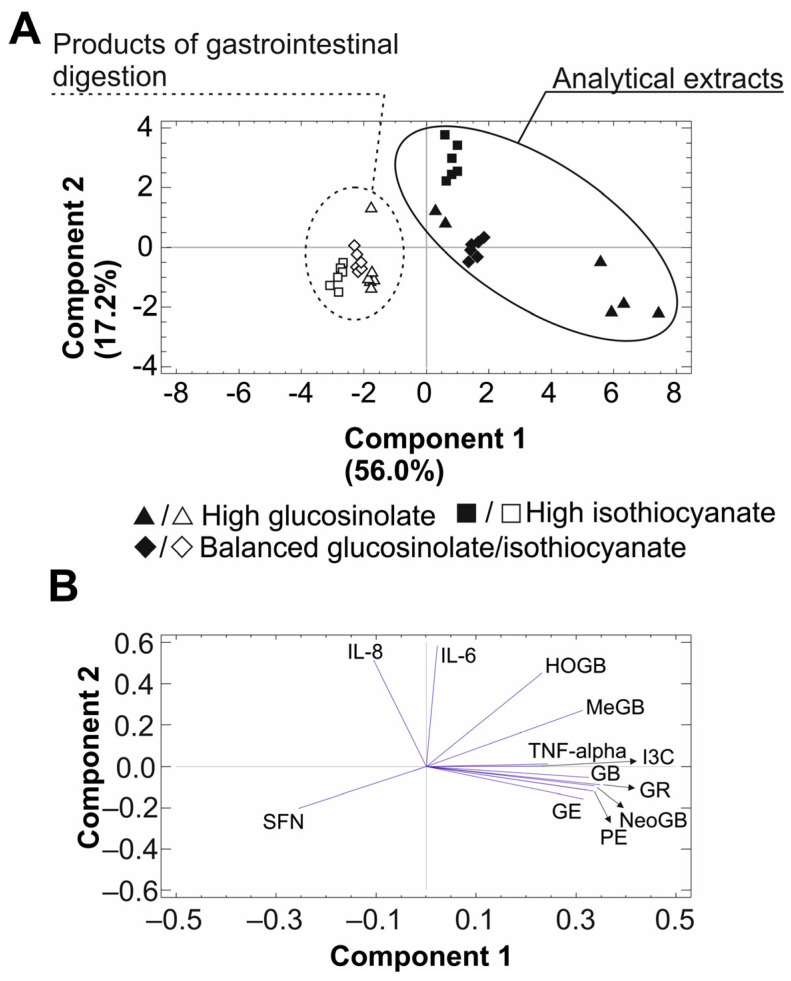
Principal Component Analysis (PCA) of the broccoli sources (analytical and bioaccessible) of bioactive organosulfur compounds. The PCA dot plot over the scores with indication of the variance explained by each separate component (**A**) the loadings biplot (**B**).

**Table 1 ijms-25-00800-t001:** Content (mg/kg dry weight (dw)) of individual glucosinolates (GSLs), and breakdown products (isothiocyanates (ITC) and nitriles) of the broccoli stalk processed to obtain high-GSL, high-isothiocyanate (ITC), and GSL/ITC balanced materials.

Analyte	High GSL Content	High ITC Content	GSL/ITC Balanced Content	LSD (*p* < 0.05)	*p*-Value ^X^
Glucosinolates					
Glucoiberin (GI)	1347.44 b	484.83 a	241.46 a	452.05	***
Glucoraphanin (GR)	7065.18 b	2423.20 a	2021.38 a	3578.07	*
Glucoerucin (GE)	1108.12 b	235.75 a	134.40 a	595.70	**
Hydroxyglucobrassicin (HGB)	59.86 a	107.90 b	66.95 a	8.08	***
Glucobrassicin (GB)	271.35 a	111.54 a	193.26 a	172.28	N.s.
Methoxyglucobrassicin (MGB)	386.05 ab	432.97 b	279.09 a	94.70	*
Neoglucobrassicin (NGB)	385.30 b	98.84 a	497.12 b	233.76	**
Gluconasturtiin (PE)	1673.18 b	513.68 a	559.63 a	1020.15	**
Isothiocyanates and indoles					
Sulforaphane (SFN)	2.72 a	4.77 a	12.31 b	0.31	***
Indole-3-carbinol (I3C)	0.16 a	0.72 b	0.92 b	1.67	***

Data are shown as means (with the indication of the least significant difference (LSD)) (*n* = 6) within the same row followed by the different lowercase letters which are significantly different at *p* < 0.05 according to the one-way analysis of variance (ANOVA) and Tukey’s multiple range test. ^X^ Significant at differences at *p* < 0.05 (*); *p* < 0.01 (**); and *p* < 0.001 (***). N.s., not significant.

## Data Availability

Data are contained within the article and Appendix A.

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
