# Peer review of "Bioaccessible Organosulfur Compounds in Broccoli Stalks Modulate the Inflammatory Mediators Involved in Inflammatory Bowel Disease"

_ijms, 2024, doi:10.3390/ijms25020800_

Round 1

Reviewer 1 Report

Comments and Suggestions for Authors

Dear Editor and Authors,

I am honored to be invited to review this manuscript. The authors presented an article, “Bioaccessible organosulfur compounds of broccoli stalks modulate the inflammatory mediators involved in intestinal bowel disease”. The topic is attractive, yet the manuscript requires modifications and implementation before being considered for publication. My requests are listed below, one by one.

Generally,

1. The manuscript mentions the regulation of intestinal diseases, but it doesn't take into account the effects of gut microbiota. This is a really important factor that must be addressed.

2. Despite the biological activity of these compounds mentioned by the authors, more specific dosages should have been given, which would have assisted in referencing other subsequent studies.

3. Appendix Amentioned by the author in the text, which is not seen in the manuscript. Tables A1-A3 are required to be included at the end of the manuscript.

Detailed

L7-10: Remove links to e-mail addresses.

L23: “IL” The first appearances should have a full name and then be abbreviated after that.

L24-26: “Most of the tested processing conditions provided plant material with significant anti-inflammatory activity and the absence of cytotoxic effects. These data confirm that SFN from broccoli stalks, processed to optimize the bioaccessible concentration of SFN.” The specific and brief results should be provided. Please rewrite these two sentences.

L103: “(GR, GE, and GI), (PE), HGB, GB, MGB, and NGB.” These abbreviations also appear for the first time. Please correct this throughout the text.

L165: “dw” These abbreviations also appear for the first time. Please correct this throughout the text.

L174: The units used the "%" are inappropriate; please convert them to µg/mL.

Figure 2: The top half of the figure has been cropped. Meanwhile, is the figure shown as a triple duplicate?

L229: “(p>0.05)” It seems that the indication provided is unnecessary. Therefore, kindly remove it along with any subsequent occurrences.

L301-302: The current description is unsuitable as it implies that food has a direct therapeutic effect, which is inaccurate. Therefore, it would be beneficial to revise the description to ensure it is factually correct and appropriate for the intended audience.

L320-323 and L338-372: It is recommended that the discussion should encompass the effects of the gut microbiota. This is an important aspect that should be taken into consideration while discussing the matter at hand.

L411, L474, and L483: “hours” should be h. Please check the full text for consistency.

L425 and L484: “4000 rpm” should be converted to Centrifugal force g.

L426 and L484: “-” should be -.

L433-435: Detailed analytical conditions are required, including column profile, temperature, and flow rate.

L455: “®” superscript.

L462: “X” should be ×

L471: same as L174

L484: “minutes” should be min. Please check the full text for consistency. L489: “n” italicized.

L503: “vs” should be and.

L505: “(surrogate physiological conditions)” That's not necessary. In vitro covers it.

L515-521: It is important to consider the impact of gut flora.

Based on the above, the manuscript only qualifies for publication in the International Journal of Molecular Sciences once the authors have improved it due to the apparent issues with the manuscript. I hope the authors will consider my recommendations to improve the manuscript.

Author Response

POINT BY POINT RESPONSE TO REVIEWER 1

  1. The manuscript mentions the regulation of intestinal diseases, but it doesn't take into account the effects of gut microbiota. This is a really important factor that must be addressed.

According to the reviewer’s suggestion, the relevance of gut microbiota in intestinal inflammation was referred to in the introduction (page 2, lines 49-57 of the reviewed version of the manuscript (MS)).

  1. Despite the biological activity of these compounds mentioned by the authors, more specific dosages should have been given, which would have assisted in referencing other subsequent studies.

According to the reviewer’s comment and additional comments below, the concentration of organosulfur compounds (both GSL and ITC) corresponding to the different supplementation ratios was provided in the material and methods subsection and the specific results subsection where the cytotoxicity of the extracts analysed was assessed.

  1. Appendix Amentioned by the author in the text, which is not seen in the manuscript. Tables A1-A3 are required to be included at the end of the manuscript.

Following the reviewer’s suggestion, the supplemental material was included at the end of the MS, on pages 14 and 15.

L7-10: Remove links to e-mail addresses.

Following the reviewer’s suggestions, the links corresponding to the e-mail addresses were removed on page 1, lines 6-10 of the reviewed version of the MS.

L23: “IL” The first appearances should have a full name and then be abbreviated after that.

According to the reviewer’s suggestion, IL was defined on page 1, lines 34-35 of the reviewed version of the MS.

L24-26: “Most of the tested processing conditions provided plant material with significant anti-inflammatory activity and the absence of cytotoxic effects. These data confirm that SFN from broccoli stalks, processed to optimize the bioaccessible concentration of SFN.” The specific and brief results should be provided. Please rewrite these two sentences.

Following the reviewer’s suggestion, additional details on the main results described in the article were included in the abstract (on page 1, lines 19-39 of the reviewed version of the MS).

L103: “(GR, GE, and GI), (PE), HGB, GB, MGB, and NGB.” These abbreviations also appear for the first time. Please correct this throughout the text.

Following the reviewer’s suggestion, the GSL abbreviations were defined the first time used in the main text (page 3, lines 120-123, of the reviewed version of the MS).

L165: “dw” These abbreviations also appear for the first time. Please correct this throughout the text.

Following the reviewer’s suggestion, the abbreviation dw was described for the first time used in the main text (page 4, line 133), as well as in the tables and figures captions that should stand by themselves.

Figure 2: The top half of the figure has been cropped. Meanwhile, is the figure shown as a triple duplicate?

Following the reviewer’s suggestion, Figure 2 was edited for higher resolution (page 5). Also in the figure caption it was provided information on the number of replicates analysed for the calculation of cell viability (page 5, line 201, of the reviewed version of the MS).

L174: The units used the "%" are inappropriate; please convert them to µg/mL.

Although the description of the supplementation level by indicating the percentage of raw materials’ extracts or bioaccessible fractions allows calculating the concentration of organosulfur compounds per mL concerning the viability assays, to enhance the understanding of the MS, this information was further detailed on pages 6, lines 204-216 of the reviewed version of the MS, in good agreement with the reviewer’s suggestion.

L229: “(p>0.05)” It seems that the indication provided is unnecessary. Therefore, kindly remove it along with any subsequent occurrences.

Following the reviewer’s suggestions, the indication of the significance degree in the statistical comparisons described in the main text was removed along the entire MS.

L301-302: The current description is unsuitable as it implies that food has a direct therapeutic effect, which is inaccurate. Therefore, it would be beneficial to revise the description to ensure it is factually correct and appropriate for the intended audience.

We agree with the comment. In this concern, on page 9, lines 329-333 of the reviewed version of the MS, the statement was redrafted to provide a more accurate description of the actual interest of brassica phytochemicals to prevent inflammatory processes.

L320-323 and L338-372: It is recommended that the discussion should encompass the effects of the gut microbiota. This is an important aspect that should be taken into consideration while discussing the matter at hand.

According to the reviewer’s suggestion, additional discussion on the role of microbiota on the anti-inflammatory effect of bioactive phytochemicals of diet, specially referred to GSL and their breakdown products, has been included on page 9, lines 347-359 of the reviewed version of the MS. Also, on page 10, lines 386-387, additional information on the microbiome-related mechanisms responsible for triggering IBD was further discussed.

L411, L474, and L483: “hours” should be h. Please check the full text for consistency.

Following the reviewer’s suggestion, “hours” was replaced “h” along the entire MS and was double-checked for enhanced consistency.

L425 and L484: “4000 rpm” should be converted to Centrifugal force g.

According to the reviewer’s suggestion, the centrifugal force was expressed as “g” in the reviewed version of the MS (page 12, line 469 and page 13, line 536).

L426 and L484: “-” should be -.

According to the reviewer’s suggestion, the minus sign for frozen conditions was expressed as “–” in the reviewed version of the MS (page 12, line 469 and page 13, line 536).

L433-435: Detailed analytical conditions are required, including column profile, temperature, and flow rate.

The detailed chromatographic conditions for both HPLC-PAD-ESI-MSn and UHPLC-ESI-QqQ-MS/MS are comprehensively provided in the references cited in the specific subsections (4.5 HPLC-PAD-ESI-MSn analysis of glucosinolates and 4.6 UHPLC-ESI-QqQ-MS/MS analysis of isothiocyanate and indole derivatives). See below. Therefore, since these have been properly cited, to avoid plagiarism issues, it was decided not to include the complete conditions but to provide the bibliographic references that would allow readers to replicate the analytical process.

REFERENCES:

4.5 HPLC-PAD-ESI-MSn analysis of glucosinolates

  • (Ref. 22) Abellán, Á.; Domínguez-Perles, R.; García-Viguera, C.; Moreno, D.A. Evidence on the Bioaccessibility of Glucosinolates and Breakdown Products of Cruciferous Sprouts by Simulated In Vitro Gastrointestinal Digestion. International Journal of Molecular Sciences 2021, Vol. 22, Page 11046 2021, 22, 11046, doi:10.3390/IJMS222011046.
  • (Ref. 54) Baenas, N.; Suárez-Martínez, C.; García-Viguera, C.; Moreno, D.A. Bioavailability and New Biomarkers of Cruciferous Sprouts Consumption. Food Res Int 2017, 100, 497–503, doi:10.1016/J.FOODRES.2017.07.049.

4.6 UHPLC-ESI-QqQ-MS/MS analysis of isothiocyanate and indole derivatives

  • (Ref. 22) Abellán, Á.; Domínguez-Perles, R.; García-Viguera, C.; Moreno, D.A. Evidence on the Bioaccessibility of Glucosinolates and Breakdown Products of Cruciferous Sprouts by Simulated In Vitro Gastrointestinal Digestion. International Journal of Molecular Sciences 2021, Vol. 22, Page 11046 2021, 22, 11046, doi:10.3390/IJMS222011046.
  • (Ref. 55) Baenas, N.; Moreno, D.A.; García-Viguera, C. Selecting Sprouts of Brassicaceae for Optimum Phytochemical Composition. 2012, doi:10.1021/jf302863c.
  • (Ref. 56) Dominguez-Perles, R.; Medina, S.; Moreno, D.Á.; García-Viguera, C.; Ferreres, F.; Gil-Izquierdo, Á. A New Ultra-Rapid UHPLC/MS/MS Method for Assessing Glucoraphanin and Sulforaphane Bioavailability in Human Urine. Food Chem 2014, 143, doi:10.1016/j.foodchem.2013.07.116.

L455: “®” superscript.

According to the reviewer’s suggestion, the ® was presented in the superscript form on page 12, line 499 of the reviewed version of the MS.

L462: “X” should be ×

According to the reviewer’s suggestion, the transepithelial unit was replaced from “X/cm2” to “x/cm2” on page 12, line 506 of the reviewed version of the MS.

L471: same as L174

The modification requested was included on page 13, lines 515-523 of the reviewed version of the MS.

L484: “minutes” should be min. Please check the full text for consistency. L489: “n” italicized.

According to the reviewer’s suggestion, “minutes” was replaced by “min” on page 13, line 536, and the whole MS was double-checked for enhanced consistency. Also, the “n” referred to the samples’ size was italicised along the MS.

L503: “vs” should be and.

According to the reviewer’s suggestion, “vs” was replaced by “and” in the reviewed version of the MS (page 13, line 555).

L505: “(surrogate physiological conditions)” That's not necessary. In vitro covers it.

Following the reviewer’s suggestion, “surrogate physiological conditions” were removed on page 13, line 557 of the reviewed version of the MS.

L515-521: It is important to consider the impact of gut flora.

Following the reviewer’s suggestion, additional conclusive remarks on the relevance of the microbiota and prospects on the analysis of the pre-biotic effect of bioaccessible extracts of broccoli stalks were included on page 14, lines 266-576.

Reviewer 2 Report

Comments and Suggestions for Authors

This work involves research of the anti-inflammatory activity of the bioaccessible fraction of broccoli stalks. Researchers applying different conditions to stalks examining anti inflammatory activity of the bioactive sulforaphane in the broccoli stalk. The way antiinflammatory activity is assessed is by modulation of  expression of pro-inflammatory cytokines (IL- 90 6, IL-8, and TNF-α) secreted by cells exposed to a pro-inflammatory stimulus. 

Abstract: Well presented with good review of the key essentials of the article. 

Introduction: Good background information on the specific research area in question. Provided background on the individual glucosinolates (GSL), and  breakdown products (isothiocyanates (ITC) of interest in the broccoli stalk. 

Results: Logical sequence of experiments conducted.  Table showing glucosinolate breakdown products. Concentration of bioaccessible sulforaphane. Profile of pro-inflammatory interleukins (IL-6, IL-8, and TNFα) induced by IL-1β in intestinal epithelium cells mediated by the modulatory capacity of raw material and digestive extracts of broccoli stalk processed to obtain high GSLs.  Cell viability testing.  Principal component analysis showing relationship between the digestive products and the raw material extracts. 

Discussion: From these results, it was addressed the extent to which the mixtures of GSL, ITC, and indoles of raw materials’s extracts and the bioaccessible could help to adjust the profile of relevant proinflammatory cytokines (IL-6, IL-8, and TNFα) secreted by intestinal epithelial cells exposed to a pro-inflammatory stimulus and thus, control de migration and maturation of the lamina propria resident macrophages.. Materials and Methods: Appropriate. 

Emphasis needs to be made to point out the in vitro nature of this work and the limited n of samples. This can be provided in the discussion. 

Otherwise, the manuscript is well I would vote to accept with minimal revisions. 

Author Response

POINT BY POINT RESPONSE TO REVIEWER 2

Abstract: Well presented with good review of the key essentials of the article. 

Thank you very much for the positive consideration of our research.

Introduction: Good background information on the specific research area in question. Provided background on the individual glucosinolates (GSL), and  breakdown products (isothiocyanates (ITC) of interest in the broccoli stalk. 

Thank you very much for the positive consideration of our research.

Results: Logical sequence of experiments conducted.  Table showing glucosinolate breakdown products. Concentration of bioaccessible sulforaphane. Profile of pro-inflammatory interleukins (IL-6, IL-8, and TNFα) induced by IL-1β in intestinal epithelium cells mediated by the modulatory capacity of raw material and digestive extracts of broccoli stalk processed to obtain high GSLs.  Cell viability testing.  Principal component analysis showing relationship between the digestive products and the raw material extracts. 

Thank you very much for the positive consideration of our research.

Discussion: From these results, it was addressed the extent to which the mixtures of GSL, ITC, and indoles of raw materials’s extracts and the bioaccessible could help to adjust the profile of relevant proinflammatory cytokines (IL-6, IL-8, and TNFα) secreted by intestinal epithelial cells exposed to a pro-inflammatory stimulus and thus, control de migration and maturation of the lamina propria resident macrophages.

Thank you very much for the positive consideration of our research.

Materials and Methods: Appropriate. 

Thank you very much for the positive consideration of our research.

Emphasis needs to be made to point out the in vitro nature of this work and the limited n of samples. This can be provided in the discussion. 

According to the reviewer’s suggestion, information on the limitation of the models set up in this research was provided through out the entire MS, also identifying the future prospects needed for the actual application of the new knowledge generated.
